# Impact of RBMS 3 Progression on Expression of EMT Markers

**DOI:** 10.3390/cells13181548

**Published:** 2024-09-14

**Authors:** Tomasz Górnicki, Jakub Lambrinow, Monika Mrozowska, Klaudia Krawczyńska, Natalia Staszko, Alicja Kmiecik, Aleksandra Piotrowska, Agnieszka Gomułkiewicz, Hanna Romanowicz, Beata Smolarz, Marzena Podhorska-Okołów, Jędrzej Grzegrzółka, Agnieszka Rusak, Piotr Dzięgiel

**Affiliations:** 1Division of Histology and Embryology, Department of Human Morphology and Embryology, Wroclaw Medical University, T. Chalubinskiego 6a St., 50-368 Wroclaw, Poland; klambrinow@gmail.com (J.L.); monika.mrozowska@umw.edu.pl (M.M.); klaudia.krawczynska@umw.edu.pl (K.K.); natalia.staszko@student.umw.edu.pl (N.S.); alicja.kmiecik@umw.edu.pl (A.K.); aleksandra.piotrowska@umw.edu.pl (A.P.); agnieszka.gomulkiewicz@umw.edu.pl (A.G.); jedrzej.grzegrzolka@umw.edu.pl (J.G.); agnieszka.rusak@umw.edu.pl (A.R.); piotr.dziegiel@umw.edu.pl (P.D.); 2Laboratory of Cancer Genetics, Department of Pathology, Polish Mother’s Memorial Hospital Research Institute, Rzgowska 281/289, 93-338 Lodz, Poland; hanna-romanowicz@wp.pl (H.R.); smolbea@o2.pl (B.S.); 3Division of Ultrastructure Research, Department of Human Morphology and Embryology, Wroclaw Medical University, T. Chalubinskiego 6a St., 50-368 Wroclaw, Poland; marzenna.podhorska-okolow@umw.edu.pl

**Keywords:** RBMS 3, epithelial-to-mesenchymal transition (EMT), breast cancer, therapeutic strategies, biomarkers

## Abstract

Epithelial-to-mesenchymal transition (EMT) is a complex cellular process that allows cells to change their phenotype from epithelial to mesenchymal-like. Type 3 EMT occurs during cancer progression. The aim of this study was to investigate the role of RNA-binding motif single-stranded interacting protein 3 (RBMS 3) in the process of EMT. To investigate the impact of RBMS 3 on EMT, we performed immunohistochemical (IHC) reactions on archived paraffin blocks of invasive ductal breast carcinoma (*n* = 449), allowing us to analyze the correlation in expression between RBMS 3 and common markers of EMT. The IHC results confirmed the association of RBMS 3 with EMT markers. Furthermore, we performed an in vitro study using cellular models of triple negative and HER-2-enriched breast cancer with the overexpression and silencing of RBMS 3. RT-qPCR and Western blot methods were used to detect changes at both the mRNA and protein levels. An invasion assay and confocal microscopy were used to study the migratory potential of cells depending on the RBMS 3 expression. The studies conducted suggest that RBMS 3 may potentially act as an EMT-promoting agent in the most aggressive subtype of breast cancer, triple negative breast cancer (TNBC), but as an EMT suppressor in the HER-2-enriched subtype. The results of this study indicate the complex role of RBMS 3 in regulating the EMT process and present it as a future potential target for personalized therapies and a diagnostic marker in breast cancer.

## 1. Introduction

Recent global cancer statistics indicate that breast cancer is the second most commonly diagnosed cancer worldwide, accounting for 11.6% of all new cases of cancer [1,2]. Given its prevalence, breast cancer represents not only a significant health concern but also an economic challenge [3]. Although considerable advances have been made in the diagnosis and treatment of breast cancer in recent decades, there is a pressing need for further research, particularly focused on the identification of new biomarkers and potential therapeutic targets [4]. From a molecular perspective, breast cancer can be classified into four main subtypes, luminal A, luminal B, HER-2-enriched, and triple negative breast cancer (TNBC), with TNBC being the only subtype lacking a dedicated targeted treatment [5,6]. Patients diagnosed with metastatic breast cancer generally have a poorer prognosis compared to those with non-metastatic disease [7].

One of the principal phenomena associated with metastasis is epithelial-to-mesenchymal transition (EMT) [8,9]. Some researchers suggest that EMT may even be necessary not only for tumor metastasis but also for the entire process of carcinogenesis [10]. Therefore, investigating EMT is particularly significant in the context of breast cancer. The term “epithelial-to-mesenchymal transition” was first used in 1982, and since then, this process has remained a key area of research globally [11,12]. EMT is a complex cellular process enabling cells to transition from an epithelial to a mesenchymal-like phenotype [13]. During EMT, epithelial cells typically lose the expression of E-Cadherin, a hallmark epithelial marker, while concurrently gaining the expression of mesenchymal markers such as N-Cadherin and vimentin [14]. Three distinct types of EMT have been identified: type one, occurring in embryonic development; type two, in tissue regeneration; and type three, which is involved in cancer progression [15]. The EMT process involved in cancer progression is mediated by EMT-inducing transcription factors (EMT-TFs), including the Snail family transcriptional repressor (SNAIL), Twist Family BHLH Transcription Factor (TWIST), and Zinc finger E-box-binding homeobox (ZEB) families [16,17]. The evidence suggests that EMT-TFs are associated with processes such as drug resistance or the acquisition of stemness [18].

In their studies on EMT, researchers focused on RNA-binding motif single-stranded interacting protein 3 (RBMS 3), a protein belonging to the c-myc gene single-strand binding proteins (MSSPs) [19,20]. RBMS 3 plays a significant role not only in cancer progression but also in other pathological processes [21,22,23]. Recent studies have indicated that RBMS 3 is involved in the development of various cancers, including ovarian, colorectal, lung and breast cancer [24,25,26,27,28]. A number of papers have suggested that RBMS 3 plays a significant role in the regulation of EMT. However the exact mechanism of action of this protein remains poorly understood.

The aim of this study was to evaluate the role of RBMS 3 in the EMT process in invasive ductal carcinoma (IDC) patient-derived tumors and also in functional in vitro models of HER-2-enriched and TNBC cancer cell lines.

## 2. Materials and Methods

### 2.1. Patients Cohort

The clinical material comprised 524 paraffin blocks with clinical data from patients operated on for IDC. The clinical and pathological characteristics of the patients are presented in Table 1. Additionally, 26 paraffin blocks and clinical data from cases of mastopathy were analyzed as a control for the breast cancer cases. Patients’ clinical material was obtained from the Division of Pathomorphology of the Polish Mother’s Memorial Hospital Research Institute. The experiment was performed in accordance with the ethical standards and following the approval of the Ethics Committee of Wroclaw Medical University (decision no. KB 625/2022 25 August 2022). Table 1 presents the clinical characteristics of the studied cohort of patients.

### 2.2. Immunohistochemistry

Experiments were performed on archived paraffin blocks of invasive breast ductal carcinoma (*n* = 524) and mastopathy (*n* = 27). Whole samples were processed into tissue microarrays (TMA) and histological slides stained with hematoxylin and eosin. The samples were obtained during surgical resections performed at the Institute of Polish Mother’s Memorial Health Centre in Lodz, Poland. Immunohistochemical reactions were carried out using the following antibodies: RBMS 3 (1:200, PA5-57028, Invitrogen, Thermo Fisher Scientific, Waltham, MA, USA), TWIST 1 (1:50, ab50581, Abcam, Cambridge, UK), SNAIL (1:200, 13099-1-AP, ProteinTech, Rosemont, IL, USA), E-Cadherin (RTU, IR059, Dako, Agilent Technologies, Santa Clara, CA, USA), N-Cadherin (1:50, M3613, Dako, Agilent Technologies), SLUG (1:50, sc-166476, Santa Cruz, Dallas, TX, USA) and ZEB 1 (1:100, ab203829, Abcam). A Dako Autostainer Link 48 apparatus (Dako, Glostrup, Denmark) was used to perform the immunohistochemical reactions. Reaction visualization was carried out using EnVision™ FLEX High pH reagents (Link, Glostrup, Denmark) (Dako), according to the manufacturer’s instructions [29,30]. The reactions were scored using the Remmele and Stegner immunoreactivity scale (IRS) [31]. A cytoplasmatic reaction was detected in the case of RBMS 3, E-CAD, N-CAD and the nucleocytoplasmic in SNAIL, SLUG and TWIST 1. ZEB 1 was not detected in cancer cells; this protein was present in the stroma cells of IDC.

### 2.3. Cell Lines

Two breast cancer cell lines representing HER-2-enriched and TNBC molecular types of breast cancer, respectively (SKBR-3 and MDA-MB-231), were used in our study. All cell lines were provided by ATCC (American Type Culture Collection ATCC, Old Town Manassas, VA, USA). FBS (Sigma-Aldrich, St. Louis, MO, USA) was added to all of the media at a final concentration of 10%. The cell lines were grown in 5% CO_2_ at 37 °C. The culture media used in the study included SKBR-3-McCoy’s (ATCC) and MDA-MB-231-L-15 (Lonza, Basel, Switzerland). All media contained 1% L-glutamine and penicillin-streptomycin solutions. The cells were passaged using TrypLE™ (Gibco, Thermo Fisher Scientific, Waltham, MA, USA).

### 2.4. Lentiviral Transduction of Breast Cancer Cell Lines 

#### 2.4.1. Overexpressing RBMS 3 In Vitro Model

The lentiviral particles used for transduction in order to obtain RBMS 3 overexpression were bought from OriGene (Rockville, MD, USA). A control for overexpression was conducted using cells transduced with lentiviral control particles. The cells were transfected accordingly to the producer manual. The starting cell seeding was 0.5 × 10^5^ on a 24-well plate. The MOI used in the experiment for both cell lines was 10. For increased transduction efficacy, Polybren (10 mg/mL) was used. The cells were selected for puromycin resistance for 1 week and maintained in a medium containing 0.5 μg/mL puromycin for SKBR-3 and 2.5 ug/mL for MDA-MB-231. The effectiveness of the transduction was determined on the mRNA and protein level using RT-qPCR and WB methods.

#### 2.4.2. Silencing RBMS 3 In Vitro Model

The lentiviral particles used for transduction in order to obtain RBMS 3 silencing were bought from OriGene (Rockville, MD, USA). The control for silencing was cells transduced with scrambled shRNA. The cells were transfected accordingly to the producer manual. The starting cell seeding was 0.5 × 10^5^ on a 24-well plate. The MOI used in the experiment for both cell lines was 10. For increased transduction efficacy, Polybren (10 mg/mL) was used. The cells were selected for puromycin resistance for 1 week and maintained in a medium containing 0.5 μg/mL puromycin for SKBR-3 and 2.5 ug/mL for MDA-MB-231. The effectiveness of the transduction was determined on the mRNA and protein level using RT-qPCR and WB methods.

### 2.5. Migration Assay

SKBR-3 and MDA-MB-231 cells with overexpression and silenced RBMS 3, as well as wild-type SKBR-3 and MDA-MB-231 cells, were seeded in a Culture-Insert 2 Well in µ-Dish 35 mm (Ibidi, Gräfelfing Germany). The cells were seeded at 3.5 × 10^4^ per well. After 24 h, the inserts were removed, and initial images were taken using a BX41 light microscope (Olympus, Tokyo, Japan). Twenty-four hours after the insert removal, a final set of images were taken to establish changes in the migration between types of cells.

### 2.6. Immunofluorescence Reactions

The cells were seeded at 1 × 10^4^ on the µ-Slide 8 Well high, with Glass Bottom (Ibidi, Gräfelfing Germany). The cells were fixed in cold 70% methanol. Next, the slides were washed twice in solution in PBS/0.1% Tween 20. The slides were incubated in a blocking solution of 1% BSA in solution in PBS/0.1% Tween 20 for 30 min, followed by overnight incubation with primary antibodies against E-Cadherin (1:500, sc-8226 Santa Cruz, Dallas, TX, USA) and N-Cadherin (1:500, sc-8424, Santa Cruz). The secondary antibody Goat Anti-Mouse IgG H&L (1:2000, ab150113, Abcam, Cambridge, UK) was used. The preparations were mounted in Prolong DAPI Mounting Medium (Invitrogen, Carlsbad, CA, USA). The observations were made at objective 60 × 1.40 oil, with the use of a Fluoview FV3000 confocal microscope (Olympus, RRID:SCR_017015) coupled with Cell Sense software version 2.6 (Olympus, RRID:SCR_016238). The analysis of the reaction presence and determination of the cell count presenting a reaction were performed using Ilastik Version 1.4.0.post1 (10 November 2023). All images were analyzed using the same parameters of threshold and size. The percentage of positive cells was calculated by dividing the number of positive reactions by the number of visible cell nuclei. For each group, measurements were taken at three distinct hot spots [32].

### 2.7. Real-Time PCR

Total RNA from the cell samples was isolated using the RNeasy Mini Kit (Qiagen, Hilden, Germany). cDNA was obtained with an iScript cDNA Synthesis Kit (Bio-Rad Laboratories, Hercules, CA, USA), according to the manufacturer’s protocol. A 7500 Real-Time PCR System instrument and 7500 software v2.0.6 (Applied Biosystems, Waltham, MA, USA) were applied to carry out a real-time PCR. The TaqMan probes used in the experiment were as follows: Hs01104892_m1 for RBMS3, Hs04989912_s1 for TWIST1, Hs00195591_m1 for SNAIL, Hs01023895_m1 for E-Cadherin, Hs00983056_m1 for N-Cadherin, Hs00161904_m1 for SLUG and Hs01566408_m1 for ZEB 1 (Applied Biosystems). The expression level of β-actin (Hs99999903_m1, Applied Biosystems) was used as an endogenous control for further normalization purposes. The experiments were run in triplicate. Reactions were carried out under the following conditions: polymerase activation at 50 °C for 2 min and initial denaturation at 94 °C for 10 min, followed by 40 cycles of denaturation at 94 °C for 15 s and annealing and elongation at 60 °C for 1 min. The amount of cells used for the RT-qPCR was 1.0 × 10^6^ of each type of cells. The ΔΔCt method was used to determine the relative mRNA expression.

### 2.8. Western Blotting

For protein expression analysis, 2.5 × 10^6^ cells of each type were lysed on ice using CelLytic™ MT Cell Lysis Reagent (Sigma-Aldrich, Burlington, MA, USA) with the addition of Halt™ Protease Inhibitor Cocktail 100× (Thermo Fisher Scientific, Waltham, MA, USA) and 2 mM PMSF (phenylmethylsulphonyl fluoride) (Sigma-Aldrich). The samples were then centrifuged at 12,000× *g* for 10 min at 4 °C to collect the protein-containing supernatant. The protein concentration was determined by colorimetric analysis using bicinchoninic acid (Pierce BCA Protein Assay Kit, Thermo Fisher Scientific, Waltham, MA, USA) and NanoDrop 1000 (Thermo Fisher Scientific, Waltham, MA, USA). In this study, we used total protein normalization, because the studies suggest that this method may be better than analyzing the expression of housekeeping genes [33]. We used a full stain-free system from Bio-Rad. The samples were mixed with sample loading buffer (250 mM Tris pH = 6.8, 40% glycerol, 20% (*v*/*v*) β-mercaptoethanol, 0.33 mg/mL bromophenol blue, 8% SDS) and denatured at 96 °C for 10 min. Samples containing 30 µg of protein were loaded onto Mini-Protean TGX stain-free gels (Bio-Rad) and separated by SDS-PAGE under reducing conditions. A stain-free Western blot was performed according to the supplier’s protocol (Bio-Rad). The gel was activated in order to capture stain-free gel images. Proteins were transferred to a low-fluorescence PVDF (LF-PVDF) membrane using the Trans-Blot Turbo Transfer System #1704150 (Bio-Rad). After transfer, a stain-free image of the LF-PVDF membrane was obtained. Next, the membrane was incubated in a blocking solution (5% milk in TBST) for 1 h at room temperature (RT), followed by overnight incubation at 4 °C with the following antibodies: RBMS 3 (1:2000 the same antibody used for IHC reactions), TWIST1 (1:200 the same antibody used for IHC reactions), SNAIL (1:500 the same antibody used for IHC reactions), E-Cadherin (1:200, sc-8226, Santa Cruz, Dallas, TX, USA), N-Cadherin (1:200, sc-8424, Santa Cruz), SLUG (1:100) and ZEB-1 (1:500). The membrane was then washed with TBST buffer (0.1% Tween 20 in PBS) and incubated for 1 h at RT with HRP-conjugated anti-rabbit or anti-mouse secondary antibodies, diluted 1:3000 (Jackson ImmunoResearch, Mill Valley, CA, USA). The next step was to obtain a stain-free image of LF-PVDF. Detection was performed using Immobilon Forte Western HRP (Merck KGaA, Darmstadt, Germany) chemiluminescent substrate. The Western blotting results were analyzed using the ChemiDoc MP system (Bio-Rad).

### 2.9. Statistical Analysis

All statistical analyses were conducted using Prism 10.0 (GraphPad Software). The results were considered statistically significant when *p* < 0.05. The Spearman’s Correlation Test was applied to analyze the existing correlations. The T-Student test was used to analyze the differences between the expression of RBMS 3 and the common EMT markers at the mRNA level.

## 3. Results

### 3.1. The Immunohistochemical Intensity of RBMS 3 Expression in Cancer Cells Correlates with Expression of Common EMT Markers in Invasive Ductal Breast Cancer (IDC) Tissue Samples

In order to provide a justification for studies concerning the role of RBMS 3 in EMT in breast cancer, we performed an analysis of the correlation between RBMS 3 immunohistochemical expression in cancer cells and the expression of Twist Family BHLH Transcription Factor 1 (TWIST 1), SNAIL (Snail Family Transcriptional Repressor 1), Snail Family Transcriptional Repressor 2 (SLUG), Epithelial Cadherin (E-CAD), Neural Cadherin (N-CAD) and Zinc finger E-box-binding homeobox 1 (ZEB 1) in 449 cases of breast cancer. An analysis of the IHC reaction unveiled that RBMS 3 is a protein present in cytoplasm, TWIST 1 in cytoplasm and nucleus, SNAIL in cytoplasm and SLUG in cytoplasm and nucleus. Both E-CAD and N-CAD are present in the cell membrane and cytoplasm of cancer cells. The statistical analysis of the results revealed a significant positive correlation of RBMS 3 expression with the expression of TWIST, SNAIL and N-CAD. In our sample group, we failed to identify ZEB 1 expression in breast cancer cells. An additional analysis of the transcriptomic data of breast invasive carcinoma (BRCA) from The Cancer Genome Atlas Program (TCGA) using the UALCAN tool [34,35] showed a statistically significant lower expression of RBMS 3 in both the HER-2-enriched and TNBC subtypes of breast cancer. Representative images of RBMS 3 and all the studied proteins, showing the expression pattern of proteins in the tissue, are presented in Figure 1. Representative images of all the studied proteins on a lower magnification are presented in the Appendix A. A graphical presentation of significant results can be seen in Figure 2. The results of the online database analysis are presented in Figure 3; the results of the statistical analysis performed by UALCAN are presented in the Appendix A.

### 3.2. The Immunohistochemical Intensity of RBMS 3 Expression Displays Different Correlations with Common EMT Markers Depending on the Molecular Subtype of the Tumor

For further investigation, we selected from the general pool of IDC those of luminal type A, luminal type B, triple negative (TNBC) molecular status and HER-2 enriched. The analysis of correlation in the group of 80 cases of luminal A IDC did not reveal any significant correlations between the expression of RBMS 3 and the common markers of EMT.

The analysis of 170 cases of luminal B IDC revealed statistically significant positive correlation between the expression of RBMS 3 and expression of TWIST 1, SNAIL and N-CAD. The statistically significant data are presented in Figure 4.

In the group of 37 TNBC cases, a significant positive correlation was observed between RBMS 3 and TWIST 1 in cancer cells and a borderline significant correlation with SLUG expression (*p* < 0.09). With regard to other markers, there is a visible trend that the expression of markers increases with RBMS 3 expression, but the results are not statistically significant.

In 20 cases of HER-2-enriched IDC cases, we did not observe a correlation between RBMS 3 and the expression of the studied EMT markers. The positive correlation with E-CAD was close to statistical significance (*p* = 0.053). The statistical data are presented in Figure 5.

An analysis of the IHC data in regards to histological grade unveiled that in the HER-2-enriched subtype there were significant correlation with N-CAD in grade 2. In the case of TNBC cases, we observed a significant correlation with TWIST 1 in grade 2. We also performed an analysis of the correlations between RBMS 3 and EMT markers with lymph node invasion, revealing that in the HER-2-enriched subtype there were positive correlations with TWIST 1 in tumors with lymph node metastasis, whereas in TNBC cases there were positive correlations with TWIST 1, as well as N-CAD. The results are presented in Figure 6.

### 3.3. In Vitro Models of Triple Negative and HER-2-Positive Breast Cancer Cell Lines with Overexpression of RBMS 3 Lead to Observation of Additional Specific Bands

In order to investigate the effect of RBMS 3 expression on two of the most aggressive types of breast cancer, we have developed models of the triple negative breast cancer cell line: one MDA-MB-231 with an overexpression of RBMS 3 and one with a silenced RBMS 3 using lentiviral particles. The same goes for the HER-2-enriched cell line SKBR-3. We also created two models of this cell line, one with overexpressed and one with silenced RBMS 3. Figure 7 and Figure 8 show data indicating this successful silencing and overexpression. As per our best knowledge, for the first time we show that the intense overexpression of RBMS 3 both in the MDA-MB-231 and SKBR-3 cell lines leads to the detection of two additional specific bands with a molecular weight of about 65 kDa and 80 kDa in MDA-MB-231 cells and 70 kDa and 100 kDa in SKBR-3 cells, in addition to the baseline protein with a molecular weight of 55 kDA.

### 3.4. In Vitro Model of RBMS 3 Overexpression Unveils Differences in RBMS 3 Impacts on EMT Markers between Molecular Subtypes

In the following step of the study, we examined the impact of RBMS 3 overexpression in the MDA-MB-231 cell line on the expression of EMT markers used in the immunohistochemical assessment. To elucidate changes both at the mRNA and protein levels, we used RT-qPCR and Western Blot methods. On the mRNA level, all the tested markers (TWIST 1, SNAIL, SLUG, E-CAD, N-CAD and ZEB 1) were significantly elevated. At the protein level, there was an observable increase in E-CAD, N-CAD and ZEB 1 amounts. Interestingly, the TWIST 1 protein were not observed on the standard level of 28 kDa; instead of that, we observed bands at heights of about 90 kDa and 120 kDa (Figure 9).

The SKBR-3 cell line with an overexpression of RBMS 3 demonstrated a statistically significant reduction in the expression of SLUG, N-CAD and ZEB 1 mRNA compared to the negative control. No significant changes were observed in the expression of TWIST 1, SNAIL and E-CAD mRNA. On the protein level, ZEB 1 in SKBR-3 cells were below the detection threshold. No significant changes were observed in the levels of the other proteins (Figure 10).

### 3.5. In Vitro Models with Lentiviral-Silenced RBMS 3 Expression Impact Common Markers of EMT Differently Depending on the Molecular Subtype

The silencing of RBMS 3 in triple negative breast cancer changed the expression of the studied proteins in a different way to overexpression. At the mRNA level, there is an observed significant reduction in the amount of TWIST 1, SNAIL, SLUG, and ZEB 1. The level of N-CAD does not change significantly, and in the opposition level, E-CAD significantly increased. At the protein level, there is an observable change in the expression of SLUG and E-CAD (Figure 11).

In the HER-2-enriched model of RBMS 3 silencing, there was a significant decrease in TWIST 1 and E-CAD on the mRNA level, as well as a significant increase in N-CAD and ZEB 1. On the other hand, no observable changes were observed in the protein levels, except for SLUG, where an increase in protein level was noted. This aligned with the result of the RT-qPCR, which presented an increase in SLUG mRNA level, though not a statistically significant one (Figure 12).

### 3.6. Changes in RBMS 3 Expression Impact Motility and the Localization of E-CAD and N-CAD in Studied Cells

In order to investigate the impact of changes in RBMS 3 expression on the migration of cells, we performed a scratch test. The results of the scratch test presented in Figure 13 and Figure 14 unveil that in the TNBC model, the silencing of RBMS 3 impairs the migration of cancer cells more strongly than overexpression. In the HER-2-enriched model, no changes in migration were observed between the silencing and overexpression of RBMS 3.

To further study the potential of cells to migrate, we conducted an immunofluorescent analysis of E-CAD and N-CAD expression in relation to RBMS 3 status. The results of this study on the TNBC model showed the presence of the cytoplasmatic expression of E-CAD and N-CAD in RBMS 3-overexpressing cells in contrast to the cells with silenced RBMS 3, where traces of E-CAD and N-CAD were observed in the cell’s cytoplasm. The results of the immunofluorescent staining are presented in Figure 15.

In the case of the HER-2-enriched model, we observed the presence of E-CAD in the cytoplasm and cell membrane of both overexpressing and silenced RBMS 3. Additionally, N-CAD was detected in some cells within the HER-2-enriched model. This observation is consistent with the results of the WB analysis. Images of the immunofluorescent staining are presented in Figure 16.

To further elucidate the exact amount of cells expressing E-CAD and N-CAD, we performed an analysis of the cell count with a positive reaction using a machine learning tool. The results obtained by this analysis showed that in the case of E-CAD, in both models the expression of E-CAD was detected in significantly more cells in RBMS 3-overexpressing cells than in RBMS 3-silenced ones. The same relationship was found in the case of N-CAD in the MDA-MB-231 cell line. In the SKBR-3 cell line, with the overexpression of RBMS 3 only around 20% of the general amount of cells was identified as N-CAD-positive, which may diminish the importance of this result. These observations are consistent with the results of the WB analysis.

## 4. Discussion

A growing body of evidence suggest the RBMS 3 may potentially play a role in the progression of breast cancer and the regulation of cell adhesion genes [28,36,37,38]. Early studies concerning the role of RBMS 3 in breast cancer depict it as a potential tumor suppressor protein capable of inhibiting metastasis [39,40]. A previous study conducted by our group on RBMS 3 expression in IDC revealed conclusions consistent with other studies, indicating that a higher expression of RBMS 3 correlates with a higher overall survival, although in vitro studies of RBMS 3 expression in cell lines shows the highest expression of RBMS 3 in TNBC and HER-2-enriched cell lines [36]. The findings of this study provide evidence that the function of RBMS 3 in breast cancer is more intricate than previously thought. The analysis of clinical material, without the subdivision into molecular subtypes, presents RBMS 3 as a positive regulator of EMT markers, suggesting a potential pro-metastatic role of RBMS 3 in breast cancer. In a selected population of TNBC and HER-2-enriched cases, the results indicate a smaller scale of RBMS 3 impact on EMT markers, where in TNBC, RBMS 3 significantly or almost significantly positively correlated with two markers and in the case of HER-2-enriched cell lines, it did so only with one, which may indicate a more epithelial phenotype of said cells.

In the study presenting RBSM 3 as an inhibitor of metastasis [39], Yang and co-workers employed the MCF-7 cell line as model for the molecular subtype of luminal A, which is one of the most benign types of breast cancer, with a 5-year survival rate of almost 95% [41]. In contrast, in our study we focused on the most aggressive types of breast cancer, which are the TNBC and HER-2-enriched types [41], and therefore MDA-MB-231 and SK-BR-3 cells were used for a functional in vitro model This in vitro model facilitated a more profound investigation into the role of RBMS 3 in IDC subtypes. Accordingly, the results demonstrate distinctions between the function of RBMS 3 in the EMT process between the HER-2-enriched model and the TNBC model. In conclusion, RBMS 3 in the HER-2-enriched model may potentially act as a suppressor of EMT, where the overexpression of RBMS 3 leads to a decrease in the expression of several EMT markers, which may result in a potentially more epithelial phenotype of cells, indicated by a decrease in N-CAD expression. On the other hand, the silencing of RBMS 3 has been observed to result in increase in EMT marker expression, changing the cell phenotype to a more mesenchymal state by increasing the expression of N-CAD, SNAIL, SLUG and ZEB 1. Our in vitro studies are consistent with the results obtained from patient-derived IDC tumor tissues, and moreover have clarified the orchestrating role of RBMS 3 in this breast cancer subtype.

It is crucial to recognize that the EMT process is not a binary phenomenon; rather, it encompasses a spectrum of hybrid phenotypes that express both mesenchymal and epithelial markers. This may provide a potential explanation for the observed increase in E-CAD expression in both of the supposedly more aggressive types of cells, namely TNBC with an overexpression of RBMS 3 and HER-2-enriched cell lines with silenced RBMS 3 [42,43]. The results of immunofluorescent staining may shed light on the potential influence of RBMS 3 on the invasive properties of breast cancer cells. In our observations, we did not observe changes in the cell phenotype depending on the expression of RBMS 3. The cytoplasmatic expression of E-CAD and N-CAD in the TNBC model of breast cancer may indicate the destabilization of cell junctions and a potential mesenchymal type of migration of this type of cells. In the HER-2-enriched model, the presence of E-CAD in the cell membrane may imply that this type of cell can exhibit a collective type of migration [44,45]. From this perspective, the expression of RBMS 3 may regulate breast cancer cells’ migration in different models, dependent on breast cancer subtypes, in relation to steroids and HER-2 receptor status.

In contrast, in the TNBC in vitro model, we observed the opposite effect to that seen in the HER-2-enriched model. The overexpression of RBMS 3 resulted in an increased expression of all the markers studied, including, interestingly, E-CAD. The increased expression of E-CAD may not necessarily imply a decrease in the potential to undergo EMT, as there is evidence that E-CAD is a necessary protein for metastasis in breast cancer [42] and that high levels of N-CAD promote cell motility in breast cancer, despite high levels of E-CAD expression [43]. On the other hand, the silencing of RBMS 3 resulted in a decrease in the expression of all the markers studied and increased the expression of E-CAD, suggesting a more epithelial phenotype of the cells. The results of the TNBC correlations with RBMS 3 are consistent with the findings of the studies conducted by Block CJ et.al [20]. In their in vitro and in vivo studies, the authors provided evidence that silencing RBMS 3 in TNBC cell lines significantly reduced cell invasion and migration. These findings are partially complementary with the results of the migration assay. In our TNBC in vitro model, which employs MDA-MB-231 cells, the silencing of RBMS 3 mostly decreased cell migration. However, the overexpression of RBMS 3 also affects MDA-MB-231 cell migration in comparison to the controls.

Contrarily, in the case of the HER-2-positive in vitro model, which was analyzed on SKBR-3 cells, the deregulation of RBMS 3 did not affect cell migration. This result, taken together with changes in E-Cadherin localization, may suggest a role of RBMS 3 in migration patterns, depending on the breast cancer subtypes, not strictly in cell migration abilities in a 2D model.

Among the proteins investigated in this study, the relationship between the expression of TWIST 1 and RBMS 3 expression in triple negative breast cancer seems to be the one with the most contradictory results obtained by scientists to date. In the initial study on this association by Zhu L. et al., the authors found that TWIST 1 expression is significantly decrease in the RBMS 3-overexpressing MDA-MB-231 cell line [40]. However, Block CJ et al. provided evidence that they did not observe any changes in TWIST 1 expression in their experiments in the same model [20]. The results of our study provide potential evidence that TWIST 1 is indeed positively correlated with RBMS 3 expression. Our analysis conducted with the RT-qPCR method revealed a significant increase in TWIST 1 mRNA expression in the RBMS 3 overexpression model and a significant decrease in the RBMS 3 knockdown model. In the Western blot analysis, we did not observe TWIST 1 at the standard level, but specific bands were present at a higher molecular weight, which, according to the literature, may correspond to the ubiquitinous form of TWIST 1 [46]. The same conclusions can be drawn for the SNAIL and SLUG proteins, where the amount of nominal protein was low and specific bands were present at the higher levels of the molecular mass ladder [47,48]. The positive correlation of TWIST 1 with RBSM 3 in TNBC is supported by the results of the IHC analysis. It is important to note that this correlation is true not only for TNBC cases, but also for HER-2-enriched cases and for the general population of all IDC specimens studied. The differences in the results of the 2D models may stem from the fact that the model prepared by Block CJ et.al [20] was based on a human mammary epithelial cell line rather than an actual cancer cell line. Additionally, we observed multiple bands of E-CAD and ZEB 1 in WB images. The presence of multiple bands of E-CAD may be caused by the presence of the N-terminal fragment of E-CAD, which is the product of the proteolytic degradation of E-CAD that may be present in cancer cells [49,50]. In the case of ZEB 1, the additional bands may be the products of proteasomal degradation, due to the ZEB-1 protein’s short half-life being only 3 h [51].

The high efficiency of RBMS 3 overexpression in both cell lines studied led to the generation of two additional heavier products of RBMS 3 mRNA. In both cell lines, the additional products were at the same molecular weight, suggesting that RBMS 3 post-translational modifications are not bound to the molecular type of breast cancer. These results may indicate that the overexpression of RBMS 3 may lead to the formation of, to our knowledge, previously undescribed forms that require further investigation.

Taking into consideration all the results presented in this study, we propose a novel multidimensional view of the role of RBMS 3 in EMT in breast cancer, suggesting that RBMS 3 plays different roles in orchestrating the protein expression levels involved in EMT depending on the molecular type of breast cancer. This hypothesis may have significant implications for the potential use of RBMS 3 as a diagnostic marker and target for therapy. The correlation between RBMS3 and the expression of EMT markers may indicate a potential role in predicting the metastatic potential of a tumor and may be used by physicians to determine the early transition from local to systemic cancer therapy. As a target for therapy, distinguishing between the effects of RBMS 3 expression in different molecular types of breast cancer may lead to the development of more specific drugs that provide a higher treatment efficacy compared to a general drug that supposedly works for all types of breast cancer. This type of approach aligns with the current trend in medicine towards a personalized approach to cancer treatment. The results of this study also open a new field of investigation into the exact molecular mechanisms of RBMS 3’s role in EMT and the reasons for the differences in RBMS 3’s role in different subtypes of breast cancer.

### Limitations

The present study was conducted on 449 invasive ductal carcinoma tumors; however, investigations conducted on a higher number of TNBC and HER-2-enriched cases, which belong to the most rare subtypes, may strengthen the evidence for the role of RBMS 3 in these types of cancers. Furthermore, additional analysis in an in vivo model of TNBC and HER-2 tumors is required to fully elucidate the role of RBMS 3 in metastases occurring in these types of breast tumors.

## 5. Conclusions

The studies conducted indicate for the first time the role of RBMS 3 as an orchestrating protein in the EMT process in TNBC and HER-2-enriched IDC tumors. The results of our studies suggest that RBMS 3 may act as an EMT-promoting protein in TNBC and as a suppressor of EMT in the HER-2-enriched subtype. These findings shed light on the novel role of this protein in IDC tumors and may be useful for designing targeted therapies for specific breast cancer subtypes.

## Figures and Tables

**Figure 1 cells-13-01548-f001:**
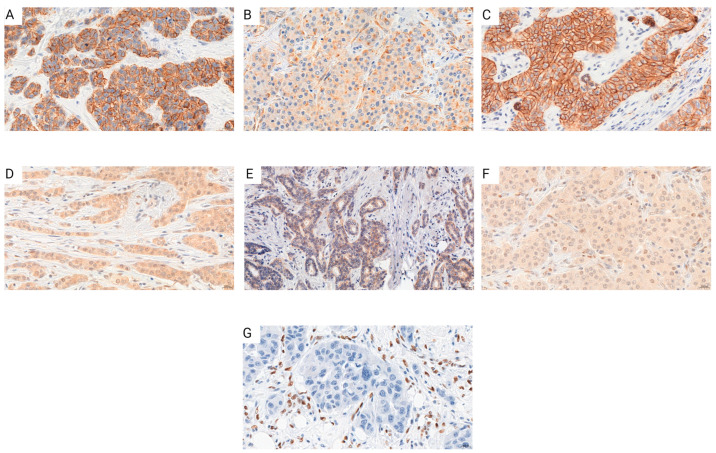
Immunohistochemical reactions performed on invasive ductal breast carcinoma tissue revealed expression of RBMS 3 and EMT markers (**A**) N-Cadherin, (**B**) RBMS 3, (**C**) E-Cadherin, (**D**) SLUG, (**E**) SNAIL, (**F**) TWIST 1, and (**G**) representation of negative ZEB 1 staining and positive in stroma of IDC. Magnification ×400. Created with BioRender.com.

**Figure 2 cells-13-01548-f002:**
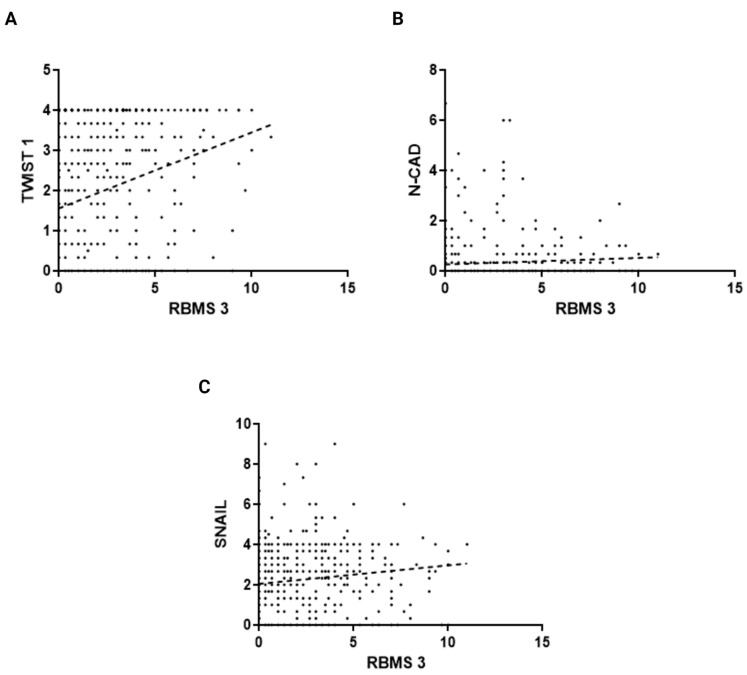
Analysis of correlation between expression of RBMS 3 in IDC with (**A**) TWIST 1 (Spearman’s Correlation Test, r = 0.31, *p* < 0.0001), (**B**) N-CAD (Spearman’s Correlation Test, r = 0.19, *p* < 0.0001) and (**C**) SNAIL (Spearman’s Correlation Test, r = 0.18, *p* < 0.0001). Created with BioRender.com.

**Figure 3 cells-13-01548-f003:**
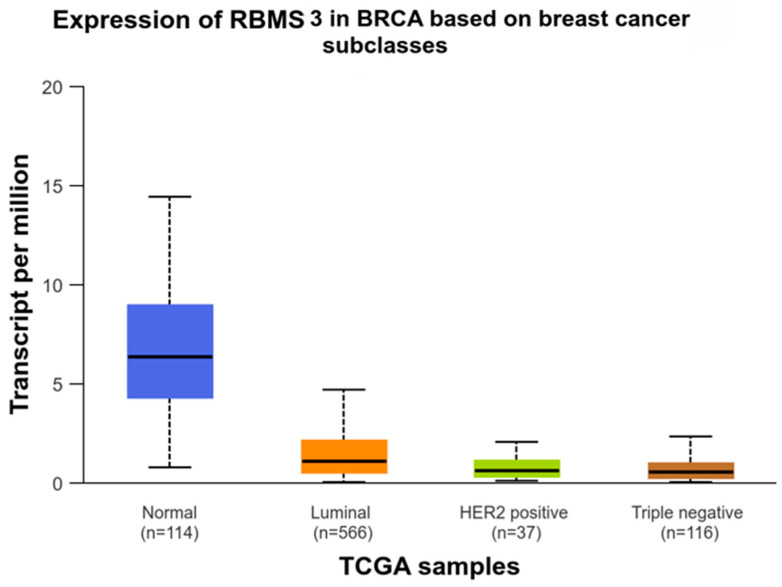
Graphical presentation of TCGA data analysis showing lower expression of RBMS 3 in both HER-2-enriched and TNBC breast cancer subtypes in comparison with healthy tissue and luminal type. Graph generated by UALCAN tool [34,35].

**Figure 4 cells-13-01548-f004:**
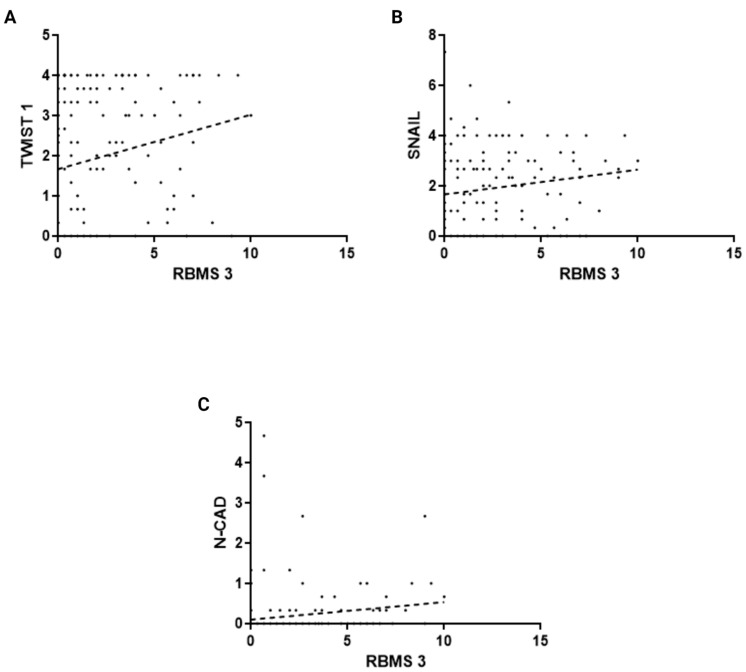
Analysis of correlation between expression of RBMS 3 in luminal B IDC with (**A**) TWIST 1 (Spearman’s Correlation Test, *p* < 0.0001, r = 0.29), (**B**) SNAIL (Spearman’s Correlation Test, *p* < 0.006, r = 0.20) and (**C**) N-CAD (Spearman’s Correlation Test, *p* < 0.0001, r = 0.30). Created with BioRender.com.

**Figure 5 cells-13-01548-f005:**
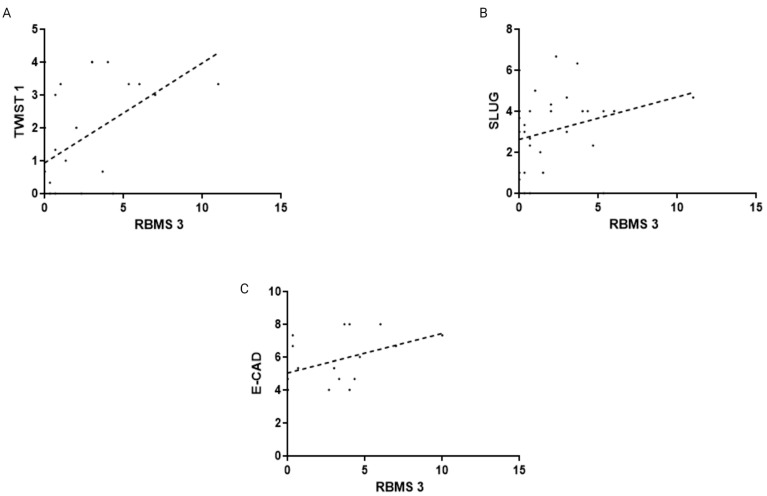
Analysis of RBMS 3 expression in TNBC IDC cases with (**A**) TWIST 1 (Spearman’s Correlation Test, *p* < 0.01, r = 0.44) and (**B**) SLUG (Spearman’s Correlation Test, *p* < 0.09, r = 0.30), and correlation between expression of RBMS 3 in HER-2-enriched IDC cases and (**C**) E-CAD (Spearman’s Correlation Test, *p* < 0.053, r = 0.48). Created with BioRender.com.

**Figure 6 cells-13-01548-f006:**
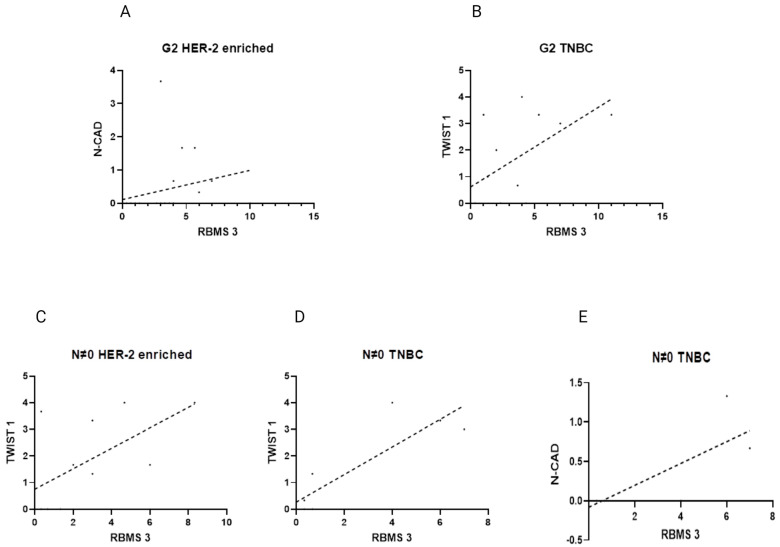
Analysis of correlation between expression of RBMS 3 and EMT makers in histological grade 2 showed positive correlations with (**A**) N-CAD in HER-2-enriched cases of IDC (Spearman’s Correlation Test, *p* < 0.003, r = 0.56) and with (**B**) TWIST 1 in TNBC cases of IDC (Spearman’s Correlation Test, *p* < 0.025, r = 0.58). Analysis of correlation between expression of RBMS 3 and EMT markers in cases with lymph node invasion showed positive correlations with (**C**) TWIST 1 in HER-2-enriched cases of IDC (Spearman’s Correlation Test, *p* < 0.035, r = 0.65) and with (**D**) TWIST 1 (Spearman’s Correlation Test, *p* < 0.031, r = 0.77) and (**E**) N-CAD (Spearman’s Correlation Test, *p* < 0.035, r = 0.85).

**Figure 7 cells-13-01548-f007:**
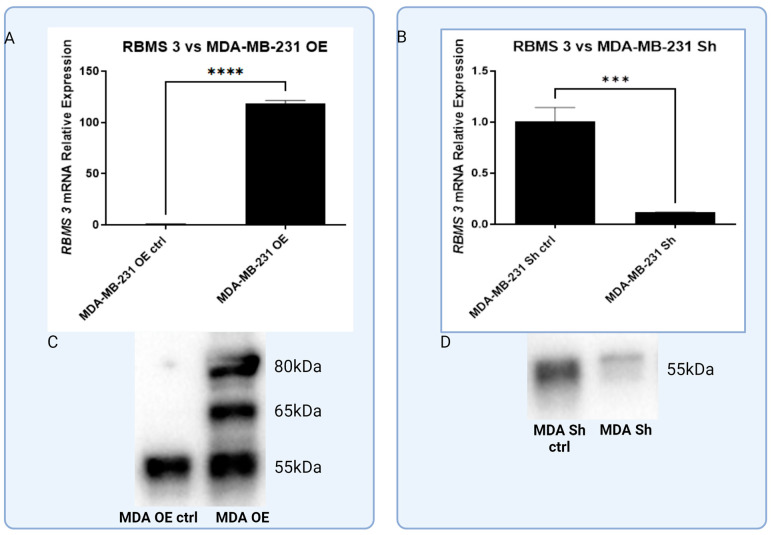
Analysis of overexpression of RBM S3 (**A**,**C**) and silencing (**B**,**D**) in MDA-MB-231 cell line on both mRNA and protein level. RT-PCR (**A**,**B**) and Western blot methods (**C**,**D**), *** *p* < 0.001, **** *p* < 0.0001. Created with BioRender.com.

**Figure 8 cells-13-01548-f008:**
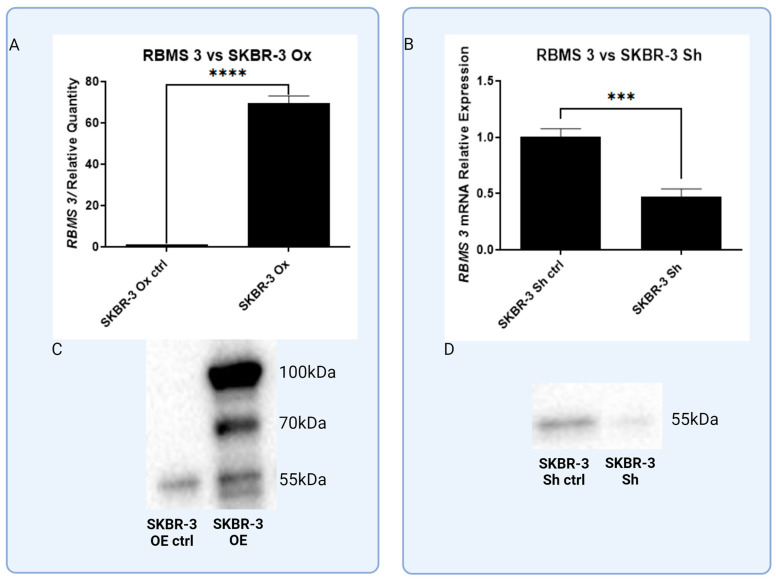
Analysis of RBMS 3 overexpression (**A**,**C**) and silencing (**B**,**D**) of RBSM 3 in SKBR-3 cell line on both mRNA and protein level. RT-PCR (**A**,**B**) and Western blot methods (**C**,**D**), *** *p* < 0.001, **** *p* < 0.0001. Created with BioRender.com.

**Figure 9 cells-13-01548-f009:**
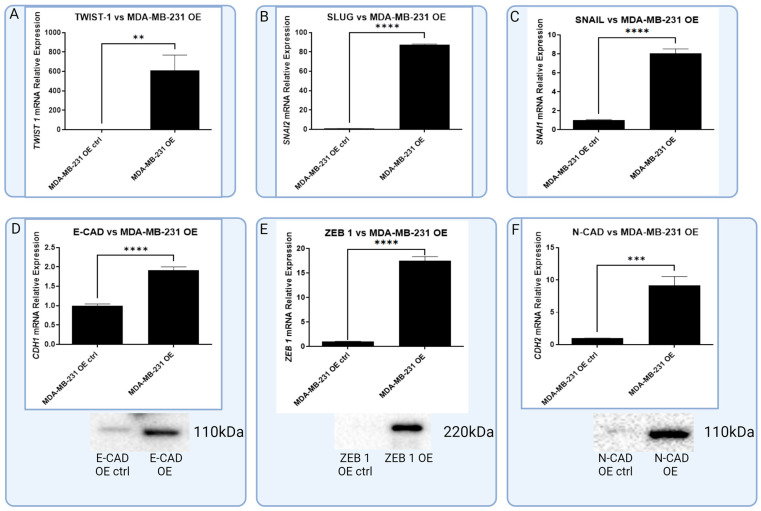
Graphic presentation of significant increase in expression of common EMT markers in RBMS 3-overexpressing TNBC cell line MDA-MB-231 on mRNA level (A–F): (**A**) TWIST 1 (T-Student test, *p* < 0.002), (**B**) SLUG (T-Student test, *p* < 0.0001), (**C**) SNAIL (T-Student test, *p* < 0.0001), (**D**) E-CAD (T-Student test, *p* < 0.0001), (**E**) ZEB 1 (T-Student test, *p* < 0.0001) and (**F**) N-CAD (T-Student test, *p* < 0.0005). Additionally in E-CAD, ZEB 1 and N-CAD (**D**–**F**), there were also observable differences in protein expression that match results of RT-qPCR, ** *p* < 0.01, *** *p* < 0.001, **** *p* < 0.0001. Created with BioRender.com.

**Figure 10 cells-13-01548-f010:**
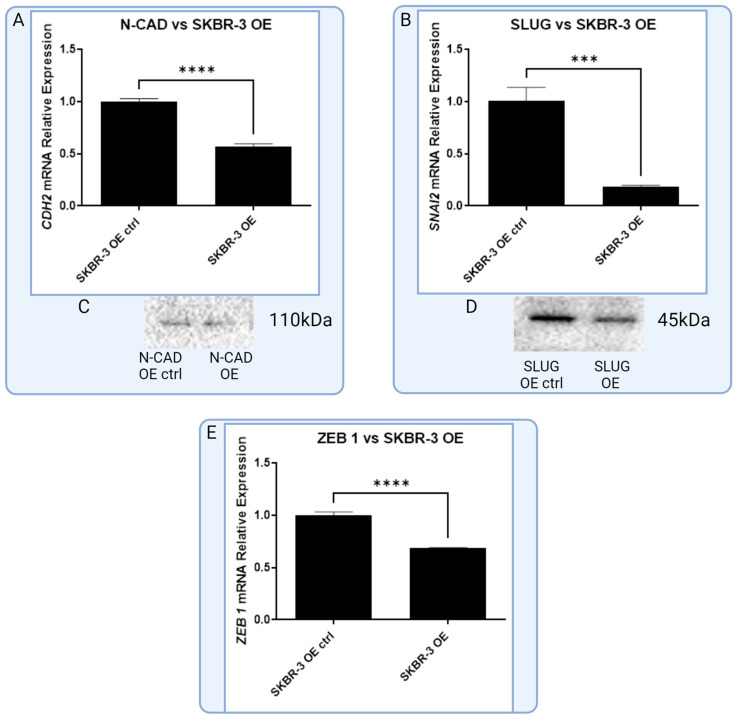
Graphic presentation of significant changes in expression of common EMT markers in HER-2-enriched cell line SKBR-3 on the level of mRNA and protein. For (**A**) N-CAD (T-Student test, *p* < 0.0001), (**B**) SLUG (T-Student test, *p* < 0.0004) and (**E**) ZEB 1 (T-Student test, *p* < 0.0001), there is statistically significant negative correlation with expression of RBMS 3. In case of N-CAD and SLUG, there are visible increases in expression of protein (**C**,**D**) in overexpressing cell line that stay in line with results of mRNA expression, *** *p* < 0.001, **** *p* < 0.0001. Created with BioRender.com.

**Figure 11 cells-13-01548-f011:**
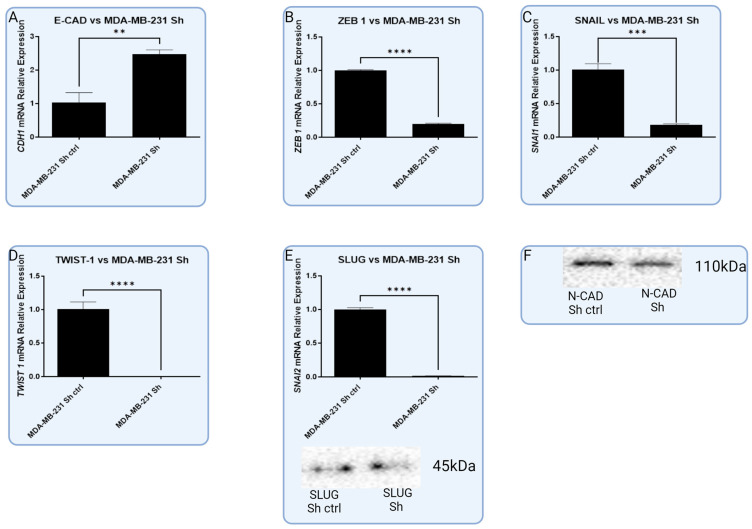
Graphic presentation of significant changes in expression of common EMT markers in RBMS 3-silenced TNBC cell line MDA-MB-231 on mRNA level (A–E) and protein level (F). Silencing of RBMS 3 lead to significant increase in expression of (**A**) E-CAD (T-Student test, *p* < 0.0001) and significant decrease in level of (**B**) ZEB 1 (T-Student test, *p* < 0.0001), (**C**) SNAIL (T-Student test, *p* < 0.0001), (**D**) TWIST 1 (T-Student test, *p* < 0.0001) and (**E**) SLUG (T-Student test, *p* < 0.0001) mRNA. In Western blot (**E**,**F**) we observed changes in (**E**) SLUG and (**F**) N-CAD protein levels between negative control and RBMS 3-silenced cells that are in line with the results of RT-qPCR, ** *p* < 0.01, *** *p* < 0.001, **** *p* < 0.0001. Created with BioRender.com.

**Figure 12 cells-13-01548-f012:**
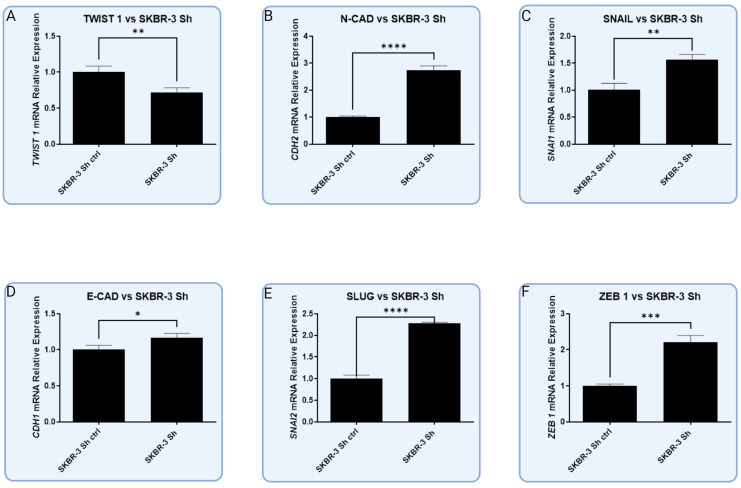
Graphic presentation of significant changes in expression of common EMT markers in HER-2-enriched cell line SKBR-3 at the level of mRNA and protein. Downregulation of RBMS 3 leads to significant decrease in level of (**A**) TWIST 1 mRNA (T-Student test, *p* < 0.009) and significant increase in all other markers: (**B**) N-CAD (T-Student test, *p* < 0.0001), (**C**) SNAIL (T-Student test, *p* < 0.003), (**D**) E-CAD (T-Student test, *p* < 0.03), (**E**) SLUG (T-Student test, *p* < 0.0001) and (**F**) ZEB 1 (T-Student test, *p* < 0.0004), * *p* < 0.05, ** *p* < 0.01, *** *p* < 0.001, **** *p* < 0.0001. Created with BioRender.com.

**Figure 13 cells-13-01548-f013:**
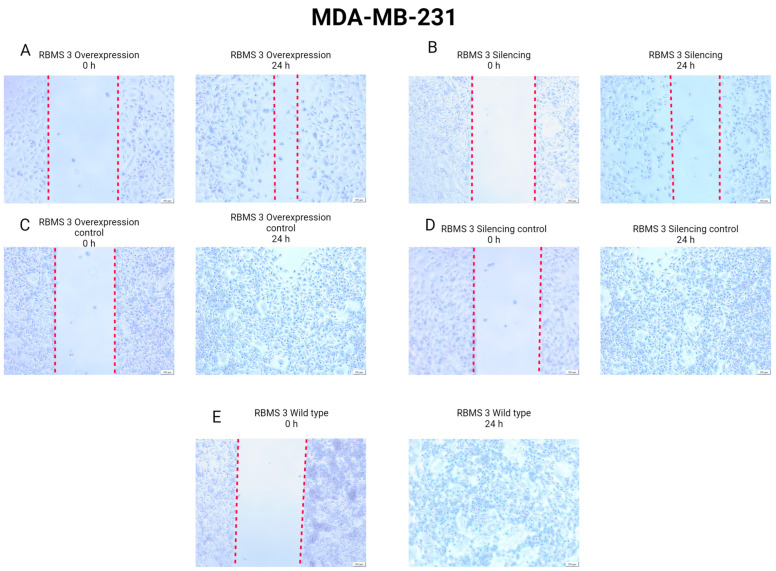
Results of scratch test performed on TNBC model (MDA-MB-231 cell line) with silenced and overexpressed RBMS 3 at the start and after 24 h. (**A**) RBMS 3 overexpression and (**C**) negative control of RBMS 3 overexpression, (**B**) RBMS 3 silencing and (**D**) negative control of RBMS 3 silencing, and (**E**) wild-type MDA-MB-231 cells. Magnification ×40. Created with BioRender.com.

**Figure 14 cells-13-01548-f014:**
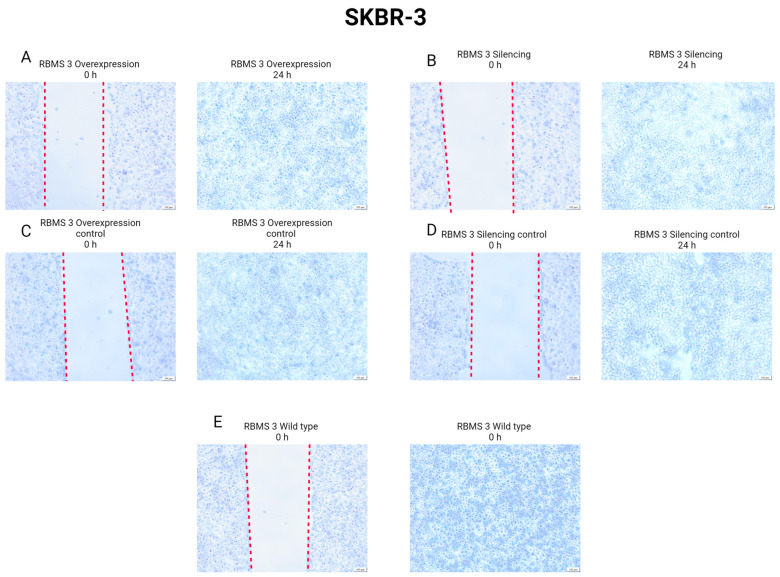
Results of scratch test performed on HER-2-enriched model (SKBR-3 cell line) with silenced and overexpressed RBMS 3 at the start and after 24 h. (**A**) RBMS 3 overexpression and (**C**) negative control of RBMS 3 overexpression, (**B**) RBMS 3 silencing and (**D**) negative control of RBMS 3 silencing, and (**E**) wild-type cells. Magnification ×40. Created with BioRender.com.

**Figure 15 cells-13-01548-f015:**
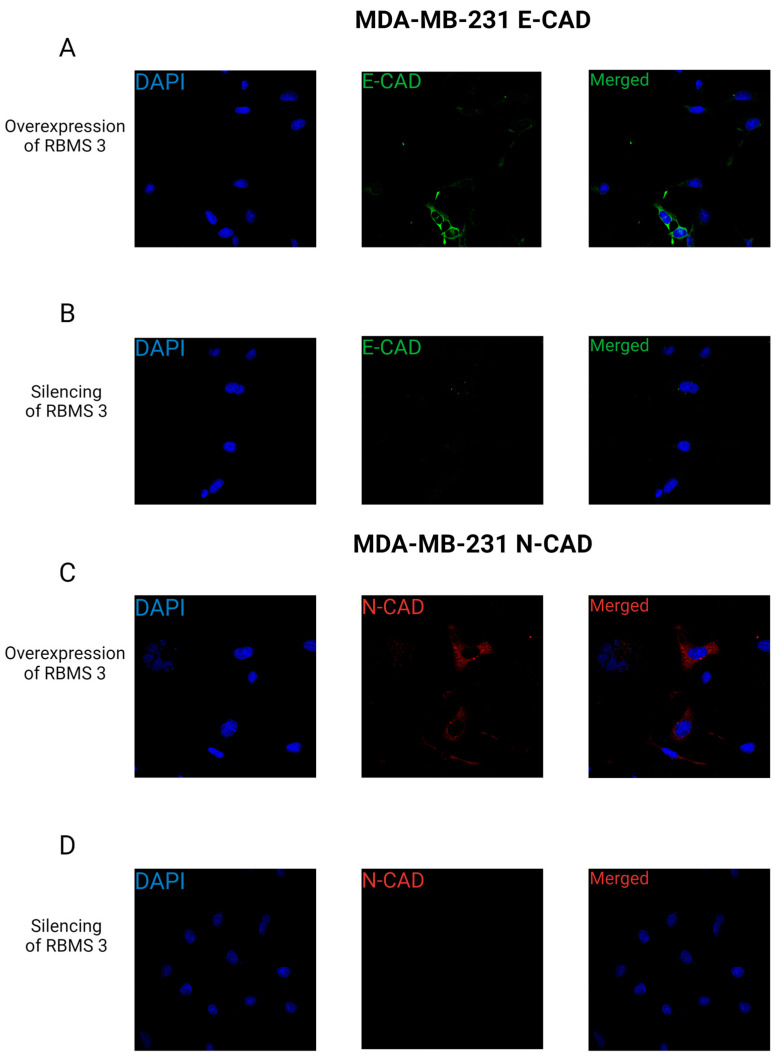
Confocal images showing expression pattern of E-CAD (**A**,**B**) and N-CAD (**C**,**D**), in MDA-MB-231 cells with silenced (**A**,**C**) and overexpressed (**B**,**D**) RBMS 3. In both cases, cytoplasmatic reactions were observed. Images were made using objective ×60. Nucleus was stained with DAPI. Created with BioRender.com.

**Figure 16 cells-13-01548-f016:**
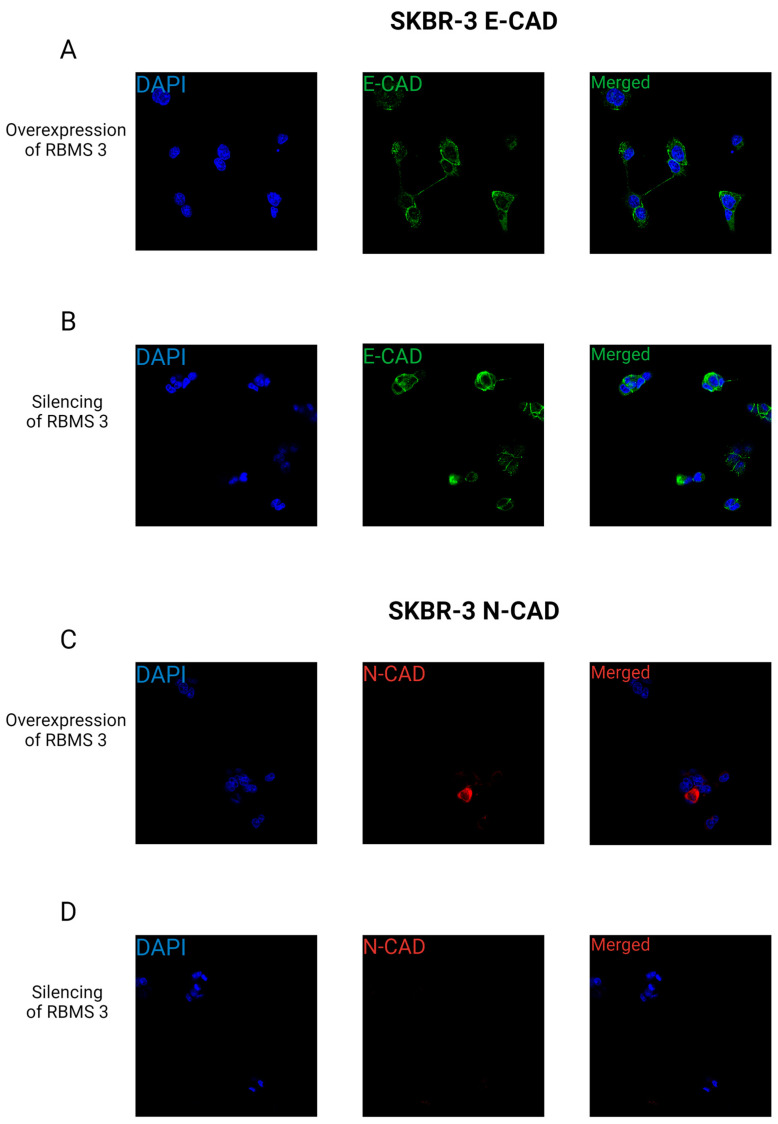
Confocal images showing membrane expression pattern of E-CAD (**A**,**B**) and N-CAD (**C**,**D**), in SKBR-3 cells with silenced (**A**,**C**) and overexpressed (**B**,**D**) RBMS 3. E-CAD expression was present in cytoplasm and cell membrane and N-CAD in cytoplasm of single cells. Images were made using objective ×60. Nucleus was stained with DAPI. Created with BioRender.com.

**Table 1 cells-13-01548-t001:** Clinical and pathological characteristics of studied patients.

Parameters	Patients	
	IHC	%
	*n* = 524	
Age		
≤60	165	31.49
>60	359	68.51
Tumor grade		
G1	87	16.60
G2	342	65.27
G3	92	17.56
unknown	3	0.57
Tumor size		
pT1	325	62.02
pT2	168	32.06
pT3	3	0.57
pT4	9	1.72
unknown	19	3.63
Lymph nodes		
pN0	314	59.92
pN1–N3	180	34.35
pNx	30	5.73
Stage		
I	224	42.75
II	257	49.05
III	18	3.44
IV	0	0
ER		
Negative	177	33.78
Positive	344	65.65
Unknown	3	0.57
PR		
Negative	183	34.92
Positive	338	64.50
Unknown	3	0.57
HER-2		
Negative	272	51.91
Positive	36	6.87
Unknown	216	51.91
Molecular tumor types		
Triple negative	37	7.06
Other types	484	92.3
Unknown	3	0.1

## Data Availability

The raw data supporting the conclusions of this article will be made available by the authors on request.

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
