# Peer review of "Impact of RBMS 3 Progression on Expression of EMT Markers"

_cells, 2024, doi:10.3390/cells13181548_

Round 1

Reviewer 1 Report

Comments and Suggestions for Authors

The topic of the manuscript and the results sound interesting. However, some issues should be addressed in order to support the results and conclusions. One of the main issues is the presentation of the results that is confuse and fragmented. To support the conclusion all the issue should be addressed.

General comments:

The entire manuscript should be reorganized and revised in order to present the introduction, the results and the relative discussion considering that EMT is not an all-or-nothing process and that hybrid phenotypes of cells can be generated during the process. These phenotypes have the concomitant expression of epithelial and mesenchymal markers, and are very aggressive and invasive.

Specific comments:

1) Abstract, line 14: change type 1 EMT to type 3 EMT.

2) Introduction, lines 42-44: EMT is not only important for tumor metastasis but for the entire process of carcinogenesis.

3) Page 6, lines 191-194: there is the legend to Figure 1, but the figure is missing.

4) Results, lines 177-184: a more detailed description of immunoreactivity and expression of EMT markers should be provided (for example if the labeling of E-cad and N-cad is at the plasma membrane or in the cytoplasm.

5) Figure 2: Insets at higher magnification should be included to show the pattern of expression of the marker.

6) Results, lines 248-255: E-cad and N-cad should be analyzed using immunofluorescence to establish if the protein expression in at the plasma membrane (cell junction integrity) or in the cytoplasm (cell junction degradation and loss of cell adhesion). This is a key point to discuss the invasive potential and the behavior of the cancer cell.

7) Figures 5-10: please include a title on each bar graph.

8) Figures 5-10 and relative description in the text: the gene expression analysis was not performed on all the genes in each experimental condition. The same for Western blot analysis: please include the Western blot results for each condition.

9) Figure 8 and relative results: why E-cad, that is a key marker of EMT, was not analyzed in SKBR-3 cells?

10) Discussion, lines 311-312: the authors suggest a complex role of RBMS 3 in breast cancer: I suggest to comment if that complexity and the effect of RBMS 3 could be different depending on the cell phenotype.

11) Discussion, lines 336-343: it is well known that carcinoma cells can be highly metastatic using different modalities to invade, the single-cell or the collective (please see Yilmaz M. et al. Trends in Mol. Med. 2007; 13:535-541, Friedl P. Curr Opin Cell Biol. 2004;16:14-23). In the second, the collective, cancer cells increase E-cad expression to improve cell adhesion and favor the collective migration. The collective migration was described for some breast cancer cells. As a consequence, the expression of E-cad must be interpreted considering that point. For these reason, most of the discussion should be revised.

Comments on the Quality of English Language

I suggest moderate editing of English language

Author Response

In accordance with the comments and suggestions received from reviewers, our manuscript (cells-3134991) has been revised.

General comments:

The entire manuscript should be reorganized and revised in order to present the introduction, the results and the relative discussion considering that EMT is not an all-or-nothing process and that hybrid phenotypes of cells can be generated during the process. These phenotypes have the concomitant expression of epithelial and mesenchymal markers, and are very aggressive and invasive.

 Thank you very much for the evaluation and comments. We reorganized and revised the manuscript according to specific comments in order to consider presence of hybrid phenotypes of cells generated during EMT process. We hope you will find this new version of the manuscript satisfactory.

Specific comments:

  • Abstract, line 14: change type 1 EMT to type 3 EMT.

Thank you very much for this valuable comment. We changed type 1 to type 3 of EMT in the following part of the abstract: “Type 3 EMT occurs during cancer progression. The aim of this study was to investigate the role of RNA binding motif single stranded interacting protein 3 (RBMS 3) in EMT.”

  • Introduction, lines 42-44: EMT is not only important for tumor metastasis but for the entire process of carcinogenesis.

Thank you very much for this valuable comment. We included this important note in the introduction: “Some researchers suggest that EMT may even be necessary not only for tumor metastasis but also for the entire process of carcinogenesis”

  • Page 6, lines 191-194: there is the legend to Figure 1, but the figure is missing.

Thank you very much for this information. We found that several parts of the submission were lost in the online submission system. We supplemented missing Figure and contacted assistant editor in order to provide you with the full display of revised manuscript.

  • Results, lines 177-184: a more detailed description of immunoreactivity and expression of EMT markers should be provided (for example if the labeling of E-cad and N-cad is at the plasma membrane or in the cytoplasm.

Thank you very much for you recommendation. We fully agreed with this, and therefore  we included a detailed description of IHC reactions in the results section: “Analysis of IHC reaction unveiled that RBMS 3 is protein present in cytoplasm, TWIST 1 in cytoplasm and nucleus, SNAIL in cytoplasm, SLUG in cytoplasm and nucleus. Both E-CAD and N-CAD are present in cell membrane and cytoplasm of cancer cells. Statistical analysis of results presented significant positive correlation of RBMS 3 expression with expression of TWIST, SNAIL and N-CAD. In our sample group we failed to identify ZEB 1 expression in breast cancer cells. Representative images of RBMS 3 and all studied proteins showing expression pattern of proteins in the tissue are presented on Figure 1. Representative images of all studied proteins on lower magnification are presented in the Supplementary materials. Graphic presentation of significant results can be seen on Figure 2.” All changes are marked in the revised version of the manuscript. We hope you will find this improved version of the manuscript satisfactory.

  • Figure 1: Insets at higher magnification should be included to show the pattern of expression of the marker.

Thank you very much for this comment, we fully agreed with this. We corrected Figure presented immunohistochemistry reactions by providing higher magnification images and  included in the Supplementary materials additional images of IHC reactions at lower magnification. We hope that you will find these changes to be satisfactory.

  • Results, lines 248-255: E-cad and N-cad should be analyzed using immunofluorescence to establish if the protein expression in at the plasma membrane (cell junction integrity) or in the cytoplasm (cell junction degradation and loss of cell adhesion). This is a key point to discuss the invasive potential and the behavior of the cancer cell.

Thank you for your suggestions. We performed immunofluorescence staining to discuss the invasive potential and behavior of the cancer cell. The description of the results was included in point 3.6 of the Results and Discussion section of the manuscript.

  • Figures 5-10: please include a title on each bar graph.

Thank you very much for your comments. We included title on each bar graph.

  • Figures 5-10 and relative description in the text: the gene expression analysis was not performed on all the genes in each experimental condition. The same for Western blot analysis: please include the Western blot results for each condition.

Thank you very much for your recommendation. We performed Western Blot and RT-qPCR in all experimental conditions. In the first version of manuscript we send all Western Blots for all experimental conditions that might have been not transferred to you, probably due to submission system failure. In the revised version we prepared file containing all supplementary materials with all raw data and also detailed results of Western Blot and RT-qPCR.

  • Figure 8 and relative results: why E-cad, that is a key marker of EMT, was not analyzed in SKBR-3 cells?

Thank you very much for this valuable comment. We analyzed the expression of E-CAD in SKBR-3 cells with overexpression of RBMS 3. However, this results of this analysis was not statistically significant. We included non-statistically significant results in the Supplementary materials attached to the manuscript.

  • Discussion, lines 311-312: the authors suggest a complex role of RBMS 3 in breast cancer: I suggest to comment if that complexity and the effect of RBMS 3 could be different depending on the cell phenotype.

Thank you very much for this important comment. We added in the discussion section: “ In our observation we did not observe changes in cell phenotype depending on the expression of RBMS 3”

11) Discussion, lines 336-343: it is well known that carcinoma cells can be highly metastatic using different modalities to invade, the single-cell or the collective (please see Yilmaz M. et al. Trends in Mol. Med. 2007; 13:535-541, Friedl P. Curr Opin Cell Biol. 2004;16:14-23). In the second, the collective, cancer cells increase E-cad expression to improve cell adhesion and favor the collective migration. The collective migration was described for some breast cancer cells. As a consequence, the expression of E-cad must be interpreted considering that point. For these reason, most of the discussion should be revised.

Thank you very much for the comment. We revised the discussion according to your suggestion and added additional paragraph in order to put emphasis on the type of migration exhibited by cells.  

I suggest moderate editing of English language.

Thank you very much for the comment, we carefully revised the manuscript in order to improve language of the manuscript. We hope that these corrections meet with your satisfaction.

Reviewer 2 Report

Comments and Suggestions for Authors

The manuscript presented by Górnicki and collaborators aims to analyze the role of RBMS3 in promoting the aggressiveness of breast cancer. To this end, they focused on the epithelial-mesenchymal transition, the main mechanism underlying the metastatic cascade. This work follows a study published in 2021, which analyzed the role and mechanism through which the same protein modulates EMT in considerable detail. The authors had already demonstrated that RBMS3 was sufficient to induce EMT in two immortalized mammary epithelial cell lines (0.1038/s41388-021-02030-x). Thus, the novelty from this perspective is quite limited. However, the findings of the different roles this protein might play in TNBC and HER2-positive cancers is interesting.

In my opinion, the manuscript lacks some essential details.

- To better correlate RBMS3 with cancer progression and thus aggressiveness, it would be useful to conduct correlation analyses between protein expression levels and patients’ clinical data, such as tumor grading, lymph node involvement and staging. Additionally, it could be valuable to investigate whether, within these categories, there is a correlation between RBMS3 and the EMT hallmarks.

- Considering that the ultimate goal of EMT is to confer invasive properties and increased migratory potential to cancer cells, it would be useful to confirm the differential role of RBMS3 in TNBC and HER2-enriched breast cancer cells subtypes through invasion/migration assays using cells overexpressing and silenced for RBMS3 in comparison to the non-transfected cells.

- Western blots cannot be accepted without a loading control. In experiments involving protein overexpression or silencing, it is also necessary to include Western blots of non-transfected cells, in addition to those transfected with the empty plasmids. Furthermore, since the focus is on the upregulation or downregulation of some proteins, I believe it is essential to present densitometric analysis as well. The Western blot of SLUG presented in Figure 9, as it stands, is not acceptable. It is evident that a technical error occurred.

- In light of the previously demonstrated role of RBMS3 in modulating TNBC cancer progression by stabilizing PRRX1 mRNA, it would be interesting and useful understand whether this mechanism is absent in HER2-positive breast cancer cell lines. This could help to hypothesize a potential mechanism underlying the dual role of this protein.

I would also add some minor concerns:

- Please make uniform the name of the proteins: i.e., RBMS3 or RMSB3.

- the Material and Methods section does not include information on the cells’ densities used for the experiments.

- In my opinion, Figure 2 should be presented as Figure 1 because it demonstrates the accuracy and reliability of the IHC, which is the basis for subsequent correlations.

- Remove all non-statistically significant correlations from the text and figures. A p-value of 0.007 or 0.009 is quite far from 0.05, especially considering the large number of samples analyzed.

- For a smoother reading, I would suggest including the name of the gene being analyzed along with the unit of measurement on the y-axis label of all the presented bar graphs, instead of using "QR" (as it is unclear what this acronym stands for).

Comments on the Quality of English Language

I suggest a revision of the manuscript to correct typos and grammar errors.

Author Response

In accordance with the comments and suggestions received from reviewers, our manuscript (cells-3134991) has been revised.

In my opinion, the manuscript lacks some essential details.

- To better correlate RBMS3 with cancer progression and thus aggressiveness, it would be useful to conduct correlation analyses between protein expression levels and patients’ clinical data, such as tumor grading, lymph node involvement and staging. Additionally, it could be valuable to investigate whether, within these categories, there is a correlation between RBMS3 and the EMT hallmarks.

Thank you very much for the valuable comment. In our previous work, cited in current manuscript  (Górnicki T, Lambrinow J, Mrozowska M, Romanowicz H, Smolarz B, Piotrowska A, GomuÅ‚kiewicz A, Podhorska-OkoÅ‚ów M, DziÄ™giel P, GrzegrzóÅ‚ka J. Expression of RBMS3 in Breast Cancer Progression. Int J Mol Sci. 2023 Feb 2;24(3):2866. doi: 10.3390/ijms24032866. PMID: 36769184; PMCID: PMC9917836.) we analyzed correlation between expression of RBMS 3 and patients clinical data. Due to small sample size of Triple negative and HER-2 enriched breast cancer cases splitting it into more groups depending on the grade, stage and lymph node invasion may lead to the distortion of results. We fully agree that this studies could be very valuable and in the future research concerning role of RBMS 3 in the EMT process in breast cancer we are going to prepare collection of more HER-2 enriched and triple negative cases in order to conduct proposed studies. At this moment, our results indicate possible mechanisms of RBMS3 and conducting results on more  We hope that this explanation will be satisfactory for you.

- Considering that the ultimate goal of EMT is to confer invasive properties and increased migratory potential to cancer cells, it would be useful to confirm the differential role of RBMS3 in TNBC and HER2-enriched breast cancer cells subtypes through invasion/migration assays using cells overexpressing and silenced for RBMS3 in comparison to the non-transfected cells.

Thank you very much for evaluation and this valuable recommendation. We performed invasion assay using cells with silenced and overexpressed RBMS 3 in comparison to non-transfected cells and recommended by producer of lentiviral particles controls. Results are presented in the section 3.6 of Results in the manuscript as well as on the Figure 11 and 12.

- Western blots cannot be accepted without a loading control. In experiments involving protein overexpression or silencing, it is also necessary to include Western blots of non-transfected cells, in addition to those transfected with the empty plasmids. Furthermore, since the focus is on the upregulation or downregulation of some proteins, I believe it is essential to present densitometric analysis as well. The Western blot of SLUG presented in Figure 9, as it stands, is not acceptable. It is evident that a technical error occurred.

Thank you very much for the comment. In this study we used stain free western blotting technique. This type of Western Blots does not require loading control. This method use total protein quantification in order to normalize results of chemiluminescence. Literature suggest that this type of Western Blot normalization is superior to the usage of housekeeping proteins (Gilda JE, Gomes AV. Stain-Free total protein staining is a superior loading control to β-actin for Western blots. Anal Biochem. 2013 Sep 15;440(2):186-8. doi: 10.1016/j.ab.2013.05.027. Epub 2013 Jun 6. PMID: 23747530; PMCID: PMC3809032.). In Supplementary materials to the manuscript we included original images of Western Blots with non-transfected cells and other not studied types of breast cancer cells. We decided to not include them in the publication due to the standard recommendation of the plasmid producer where according to the manual  any comparison in expression of genes and proteins should be compared to the cells transfected with empty plasmid as sole procedure of transduction induces significant changes in the expression of proteins in the cells. According to your recommendation performed desintometric analysis of the results and presented it in the Supplementary materials. Results of the SLUG Western Blot are the best obtainable quality Blot of this protein. We carefully conducted analysis of technical site of this Blot and obtained similar results.

- In light of the previously demonstrated role of RBMS3 in modulating TNBC cancer progression by stabilizing PRRX1 mRNA, it would be interesting and useful understand whether this mechanism is absent in HER2-positive breast cancer cell lines. This could help to hypothesize a potential mechanism underlying the dual role of this protein.

 Thank you very much for this comment. We fully agree that this idea is very interesting, worth testing and have support in the literature. Unfortunately determining exact mechanisms underlying differences between role of RBMS 3 in different types of breast cancer and connection to stabilizing PRRX1 mRNA goes beyond the scope of this manuscript as it require design of new advanced study focused on the molecular mechanisms with different sets of methods. In the future works concerning expression of RBMS 3 in breast cancer we would like to focus on this topic and include in the study design testing of your suggested hypothesis.

I would also add some minor concerns:

- Please make uniform the name of the proteins: i.e., RBMS3 or RMSB3.

Thank you very much for the comment. We carefully revised the manuscript in order to unify names of the proteins.

- the Material and Methods section does not include information on the cells’ densities used for the experiments.

Thank you very much for the comment. We included information on the cells densities used for the experiments.

- In my opinion, Figure 2 should be presented as Figure 1 because it demonstrates the accuracy and reliability of the IHC, which is the basis for subsequent correlations.

Thank you very much for the recommendation. We changed the order of figures in suggested way in order to better demonstrate the flow of the study design.

- Remove all non-statistically significant correlations from the text and figures. A p-value of 0.007 or 0.009 is quite far from 0.05, especially considering the large number of samples analyzed.

Thank you very much for the recommendation. We  removed all non-statistically significant correlations that were results of analysis of large number of samples. We would like to keep non-statistically significant correlations that apply to the results of analysis conducted on small group of  37 triple negative breast cancer samples where the small fluctuation of immunoreactive score results may significantly impact result of statistical analysis. This assumption was based on the fact that despite IHC reactions being analyzed by two independent scientists and in case of differences by the third one there is still possibility of mistake that must be count into statistical analysis.  We hope that this correction will be satisfactory for you.

- For a smoother reading, I would suggest including the name of the gene being analyzed along with the unit of measurement on the y-axis label of all the presented bar graphs, instead of using "QR" (as it is unclear what this acronym stands for).

Thank you very much for the recommendation. We included name of the gene being analyzed along with the unit of measurement being “Relative quantity”. Relative quantity indicates that we conducted analysis of the RT-qPCR with ΔΔCt method were we quantify amount of mRNA relatively to the expression of housekeeping gene in our case beta-actin. Thus we can only present results of RT-qPCR data as the percentage of housekeeping gene expression. We hope that this graphs correction will be satisfactory for you.

Round 2

Reviewer 1 Report

Comments and Suggestions for Authors

Dear Editor,

in the revised manuscript most issues were addressed and also new experiments were included. The manuscript was improved and can be accepted for publication.

Author Response

In accordance with the comments and suggestions received from reviewers, our manuscript (cells-3134991) has been revised.

In the revised manuscript most issues were addressed and also new experiments were included. The manuscript was improved and can be accepted for publication.

Thank you very much for the revision of our article and constructive criticism that substantially improved quality of our manuscript.

Reviewer 2 Report

Comments and Suggestions for Authors

I would like to thank the authors for their clarifications and the efforts to address my concerns. However, I still have some issues.

- The authors' previous publications (10.1038/s41388-021-02030-x; 10.3390/ijms24032866) clearly and strongly support the correlation of RBMS 3 with cancer progression and EMT in multiple models, including in vivo human TNBC mouse models using the MDA-MB-231 cell line, human mammary epithelial cells, TNBC cell models (MDA-MB-231 and SKBR3), and a cohort of breast cancer patients. Additionally, in the 2021 paper (10.1038/s41388-021-02030-x), they ectopically expressed RBMS3 in the HMLE cell line, resulting in the induction of a mesenchymal phenotype. They performed Western blotting, colony growth assays, and q-PCR. Subsequently, they conducted a knockout experiment of RBMS 3 in the TNBC cell lines MDAMB-231 and SUM159, analyzing the impact of protein silencing on the cells’ mesenchymal phenotype. They also assessed the invasive and migratory properties of cancer cells. The same approach has been used in the present manuscript, employing the same cell models. Thus, the novelty of this latest manuscript lies in the possible dual role of RBMS3 as an inducer of TNBC progression and as an inhibitor of breast cancer progression in HER2-enriched cancer subtypes. Although in their previous work (10.3390/ijms24032866), they actually stated that 'there were no statistically significant differences in the expression of RBMS 3 with regard to the grade, TNM, and stage of the cancer', I still believe that the correlation analysis of the RBSM 3 protein levels, revealed by IHC, with EMT hallmarks in the different G phases, stages, and TNM classification of breast cancer, independently from being TNBC or HER2 enriched subtype, could be extremely supportive of their thesis. Additionally, it could also be useful to utilize online databases to address this question concerning the expression of RBMS 3 during cancer progression in different subtypes of breast cancer.

- In my opinion, the authors should change the title of the paragraph “Changes in RBMS3 expression impacts invasive features of studied cells,” as the authors did not perform an invasion assay (e.g., Boyden chambers coated with Matrigel or extracellular matrix components, Invadopodia assay). Instead, they analyzed TNBC cell motility and the localization of E-cad and N-cad to indirectly assess TNBC aggressiveness. The scratch test does not analyze cancer cell invasiveness but rather cancer cell migration. Please correct this in the text. Additionally, the figure should be arranged more clearly, highlighting the differences in wound closure in the different experimental models used (as presented here: https://doi.org/10.3390/cells13161336).

- Confocal images should be presented as single channels (DAPI and green) and then in the merged form, including the scale bar, to better appreciate protein localization. The confocal images obtained from MDAMB-231 and SK-BR-3 should be arranged in a single figure, adding the name of the cell line corresponding to each image. Please add the name of the protein investigated with the corresponding color in the figure (as presented here: https://doi.org/10.3390/cells13110947). The number of cells presented in the four panels of Figure 13 varies significantly. Please, choose fields presenting roughly the same number of cells. Please increase the intensity of the green channel. The difference between the two breast cancer cell lines is not clearly evident. Is it possible to perform a statistical analysis on the number of cells showing cytoplasmic/nuclear protein localization in cells silenced for or overexpressing RBMS 3?

- The densitometric analysis of the Western blot (WB) bands presented in the manuscript and normalized for total protein content, should be presented together with the WB images, including a statistical analysis demonstrating effective modulation. This implies that the experiments have been repeated at least three times. The authors could organize the images showing both the mRNA and protein levels together, in a clearer manner.

- The y-axis of the graph reporting mRNA levels of different genes should be labeled as “CDH2 mRNA relative expression,” for example.

- "Ox" is usually the abbreviation for oxidation; I suggest the authors change the abbreviation for overexpression.

- Could the authors explain the multiple bands found in the original Western blot for E-cad and ZEB1?

Author Response

In accordance with the comments and suggestions received from reviewers, our manuscript (cells-3134991) has been revised.

I would like to thank the authors for their clarifications and the efforts to address my concerns. However, I still have some issues.

We would like to thank you very much for your valuable and detailed comments. We agree that suggested changes  were required in order to improve quality of this manuscript.

- The authors' previous publications (10.1038/s41388-021-02030-x; 10.3390/ijms24032866) clearly and strongly support the correlation of RBMS 3 with cancer progression and EMT in multiple models, including in vivo human TNBC mouse models using the MDA-MB-231 cell line, human mammary epithelial cells, TNBC cell models (MDA-MB-231 and SKBR3), and a cohort of breast cancer patients. Additionally, in the 2021 paper (10.1038/s41388-021-02030-x), they ectopically expressed RBMS3 in the HMLE cell line, resulting in the induction of a mesenchymal phenotype. They performed Western blotting, colony growth assays, and q-PCR. Subsequently, they conducted a knockout experiment of RBMS 3 in the TNBC cell lines MDAMB-231 and SUM159, analyzing the impact of protein silencing on the cells’ mesenchymal phenotype. They also assessed the invasive and migratory properties of cancer cells. The same approach has been used in the present manuscript, employing the same cell models. Thus, the novelty of this latest manuscript lies in the possible dual role of RBMS3 as an inducer of TNBC progression and as an inhibitor of breast cancer progression in HER2-enriched cancer subtypes. Although in their previous work (10.3390/ijms24032866), they actually stated that 'there were no statistically significant differences in the expression of RBMS 3 with regard to the grade, TNM, and stage of the cancer', I still believe that the correlation analysis of the RBSM 3 protein levels, revealed by IHC, with EMT hallmarks in the different G phases, stages, and TNM classification of breast cancer, independently from being TNBC or HER2 enriched subtype, could be extremely supportive of their thesis. Additionally, it could also be useful to utilize online databases to address this question concerning the expression of RBMS 3 during cancer progression in different subtypes of breast cancer.

Thank you for the valuable comment, we have performed an analysis of the correlation of RBMS 3 with EMT features in our database. The results of this analysis are presented in section 3.2 of the manuscript: "Analysis of IHC data in terms of histological grade showed that in the HER-2 enriched subtype there was a significant correlation with N-CAD in grade 2. In TNBC cases, we observed a significant correlation with TWIST 1 in grade 2. We also performed analysis of correlations between RBMS 3 and EMT markers with lymph node invasion and showed that in HER-2 enriched subtype there was positive correlation with TWIST 1 in tumors with lymph node metastasis, whereas in TNBC cases there was positive correlation with TWIST 1 as well as N-CAD. The results are shown in Figure 5.

In addition, we also used the online software UALCAN to analyze the TCGA transcriptomic data for the relationship between RBMS3 expression and breast cancer subtype. The results are presented in section 3.1 of the manuscript: "Additional analysis of breast invasive carcinoma (BRCA) transcriptomic data from The Cancer Genome Atlas Program (TCGA) using the UALCAN tool [32,33], showed statistically significant lower expression of RBMS 3 in both HER-2 enriched and TNBC subtypes of breast cancer". We hope this answer will be satisfactory.

- In my opinion, the authors should change the title of the paragraph “Changes in RBMS3 expression impacts invasive features of studied cells,” as the authors did not perform an invasion assay (e.g., Boyden chambers coated with Matrigel or extracellular matrix components, Invadopodia assay). Instead, they analyzed TNBC cell motility and the localization of E-cad and N-cad to indirectly assess TNBC aggressiveness. The scratch test does not analyze cancer cell invasiveness but rather cancer cell migration. Please correct this in the text. Additionally, the figure should be arranged more clearly, highlighting the differences in wound closure in the different experimental models used (as presented here: https://doi.org/10.3390/cells13161336).

Thank you very much for the valuable comment we changed the title of the requested paragraph to: “Changes in RBMS 3 expression impacts motility and the localization of E-CAD and N-CAD in studied cells”. We also corrected in terminology in the text to properly define studied properties of the cells. We also arranged the figures 13 and 14 accordingly to your suggestions.

- Confocal images should be presented as single channels (DAPI and green) and then in the merged form, including the scale bar, to better appreciate protein localization. The confocal images obtained from MDAMB-231 and SK-BR-3 should be arranged in a single figure, adding the name of the cell line corresponding to each image. Please add the name of the protein investigated with the corresponding color in the figure (as presented here: https://doi.org/10.3390/cells13110947). The number of cells presented in the four panels of

Figure 13 varies significantly. Please, choose fields presenting roughly the same number of cells. Please increase the intensity of the green channel. The difference between the two breast cancer cell lines is not clearly evident. Is it possible to perform a statistical analysis on the number of cells showing cytoplasmic/nuclear protein localization in cells silenced for or overexpressing RBMS 3?

Thank you very much for the evaluation and recommendation. We prepared new Figures 15 and 16 according to the recommendations and performed statistical analysis of cells with positive expression (Figure 17) of E-CAD and N-CAD in models with silenced and overexpressing RBMS 3 using Ilastik machine learning tool.

- The densitometric analysis of the Western blot (WB) bands presented in the manuscript and normalized for total protein content, should be presented together with the WB images, including a statistical analysis demonstrating effective modulation. This implies that the experiments have been repeated at least three times. The authors could organize the images showing both the mRNA and protein levels together, in a clearer manner.

Thank you for your valuable comment. In this manuscript, we used new stain-free methods according to Bio-Rad's instructions, which can replace the standard classical use of loading controls such as the popular β-tubulin, β-actin or GAPDH, which are common housekeeping proteins. This method has been described by Diller T et .al referenced in our manuscript under number 33.  Stain-free methods analyze total protein instead of typical loading control proteins. This technique avoids the rare cases where the level of house-keeping proteins may vary between cell lines, for example as a result of gene manipulation. In this manuscript, we have replaced the typical loading control with the stain-free technique and provided normalization according to this technique. In addition, in this manuscript we present Western blots in a more qualitative way and for this reason statistical analysis of triplicates of these results has not been provided. When using the stain-free method, the analysis is performed using the whole lane as reference. For a clear presentation of the results, we have presented the stain-free results in the Supplementary Material. We have organized Figures 7-12 according to the recommendations to show mRNA and protein levels together. We hope this answer will be satisfactory

-The y-axis of the graph reporting mRNA levels of different genes should be labeled as “CDH2 mRNA relative expression,” for example.

Thank you very much for the valuable comment we labeled the y-axis of the mRNA reporting graphs according to the recommendation.

- "Ox" is usually the abbreviation for oxidation; I suggest the authors change the abbreviation for overexpression.

Thank you very much for the valuable comment we changed the abbreviation accordingly to the recommendation to OE.

- Could the authors explain the multiple bands found in the original Western blot for E-cad and ZEB1?

Thank you very much for the valuable comment. We elaborated this topic in the discussion section of the manuscript: “ Additionally, we observed multiple bands of E-CAD and ZEB 1 in WB images Presence of multiples bands of E-CAD may be caused by presence of N-terminal fragment of E-CAD that is the product of proteolytic degradation of E-CAD that may be present in cancer cells [50,51]. In case of ZEB 1 additional bands may be the products of proteasomal degradation due to ZEB-1 protein short half-life being only 3 hours [52].” We hope this answer will be satisfactory.

Round 3

Reviewer 2 Report

Comments and Suggestions for Authors

The authors have efficiently improved the manuscript. 

Minor observations:

- Please, remove the panels regarding the analysis of fluorescence intensity observed for E-cad and N-cad (figure 17). Without a comparison to untransfected cells or an analysis of the different distribution between the nucleus and cytoplasm of RBSM 3 between silenced and over expressing cells, it risks confusing the reader and leading them to believe there is an inconsistency between the results obtained through RT-qPCR/WB and immunofluorescence.

- pag. 9 line 254 the authors are probably referring to Figure 4, not Figure 3.

Comments on the Quality of English Language

Quality of English Language needs some minor revisions

Author Response

In accordance with the comments and suggestions received from reviewers, our manuscript (cells-3134991) has been revised.

The authors have efficiently improved the manuscript. 

We would like to thank you very much for your expertise and detailed revision of our manuscript.

Minor observations:

- Please, remove the panels regarding the analysis of fluorescence intensity observed for E-cad and N-cad (figure 17). Without a comparison to untransfected cells or an analysis of the different distribution between the nucleus and cytoplasm of RBSM 3 between silenced and over expressing cells, it risks confusing the reader and leading them to believe there is an inconsistency between the results obtained through RT-qPCR/WB and immunofluorescence.

Thank you for your evaluation and recommendation. According to the recommendations we removed Figure 17 from the manuscript

- pag. 9 line 254 the authors are probably referring to Figure 4, not Figure 3.

Thank you for the comment we corrected the reference for text to refer to appropriate Figure.

Comments on the Quality of English Language

Quality of English Language needs some minor revisions

Thank you for you evaluation and recommendation we carefully revised the manuscript in order to improve English language.
